Predictive role of systemic immune-inflammation index in the prognosis of patients with advanced left-sided colorectal cancer: a retrospective study

Su Jingyue 1
Yu Shaokun 2
Xu Yanjie 1
Zhao Jiemin 1
Hu Wenwei 1
Ni Xuefeng xfnczyy@163.com 1
1 Department of Oncology, The Third Affiliated Hospital of Soochow University , Changzhou , Jiangsu , China
2 Suzhou Cancer Center Core Laboratory, The Affiliated Suzhou Hospital of Nanjing Medical University, Suzhou Municipal Hospital, Gusu School, Nanjing Medical University , Suzhou , Jiangsu , China
Connor Mark
Electronic publication date: 2025 Oct 6
Publication date: 2025
Volume: 13
Electronic Location ID: e20095
Received 2025 Jan 13; Accepted 2025 Aug 27
Copyright: ©2025 Su et al.
Copyright year: 2025
Copyright holder: Su et al.
License: This is an open access article distributed under the terms of the Creative Commons Attribution License, which permits unrestricted use, distribution, reproduction and adaptation in any medium and for any purpose provided that it is properly attributed. For attribution, the original author(s), title, publication source (PeerJ) and either DOI or URL of the article must be cited.
License URL: https://creativecommons.org/licenses/by/4.0/

Keywords: Colorectal cancer, Systemic immune-inflammation index, Chemotherapy, Prognosis

Funding: Changzhou Health and Family Planning Commission in 2018 ZD201802 This study was funded by Major Science and Technology Projects of Changzhou Health and Family Planning Commission in 2018, Grant/Award Number: ZD201802. The funders had no role in study design, data collection and analysis, decision to publish, or preparation of the manuscript.

==============================
Purpose

This study aimed to evaluate the prognostic value of the systemic immune-inflammation index (SII) in patients with advanced left-sided colorectal cancer (CRC) receiving CAPEOX ± bevacizumab as first-line chemotherapy.

Methods

A total of 231 patients with advanced left-sided CRC who received first-line CAPEOX ± bevacizumab therapy were included. Patients’ blood cell counts, clinical, and pathological data were collected before treatment, and systemic inflammatory indices were calculated, including neutrophil-to-lymphocyte ratio (NLR), platelet-to-lymphocyte ratio (PLR), monocyte-to-lymphocyte ratio (MLR), and SII. Optimal cutoffs for NLR, PLR, MLR, SII, and age were determined using X-Tile software, categorizing patients in to high- or low-value groups; clinicopathological characteristics were then compared between the high- and low-value groups within each systemic inflammatory index using chi-square tests. Survival curves were estimated using the Kaplan–Meier method, with log-rank tests applied to compare differences between groups. The Shapiro–Wilk test for normality and Spearman correlation analysis were used to evaluate the correlations among SII, NLR, PLR, and MLR. Univariable and multivariable analyses were performed with the Cox proportional hazards regression model. The prognostic value of systemic inflammatory indices was compared using the concordance index (C-index) and 5-fold cross-validation. The stability and generalizability of the C-index under varying data partitions were evaluated using mean square error (MSE). Interaction effects between treatment regimens and SII were further explored using multivariable Cox regression analysis.

Results

Univariable and multivariable Cox regression analyses identified age, primary tumor resection, SII, NLR, PLR, and MLR as independent prognostic factors for overall survival (OS). Comparative analysis of the C-index and MSE in the training and validation datasets demonstrated that SII outperformed NLR, PLR, and MLR, exhibiting the highest average C-index and the lowest MSE across 5-fold cross-validation. Patients with elevated pre-treatment SII had significantly worse OS compared to those with lower values.

Conclusion

SII is a robust prognostic marker in patients with advanced left-sided CRC receiving CAPEOX ± bevacizumab as first-line chemotherapy, demonstrating superior prognostic value compared to NLR, PLR, and MLR. Higher pre-treatment SII values were associated with worse OS, underscoring its clinical utility in prognostic stratification.

Introduction

Colorectal cancer (CRC) is a major global public health concern. According to the latest GLOBOCAN 2022 data, CRC ranks first in the burden of digestive system cancers with 1.926 million new cases and 904,000 deaths (Diao et al., 2025). Owing to the absence of early symptoms and a hesitation in performing colonoscopy, a considerable number of CRC patients are diagnosed at an advanced stage, with an unfavorable overall survival (OS) (Chen et al., 2017). Although in clinical practice, histological type, histological grade, and tumor-node-metastasis (TNM) staging are the most commonly used methods to predict CRC prognosis, prognostic heterogeneity still exists among patients with the same TNM stage (Zhang & Miao, 2023). For CRC, this heterogeneity may stem from differences in the primary tumor location and treatment regimens: on one hand, studies have indicated differences in tissue origin and biological characteristics between left-sided and right-sided CRC (Piawah & Venook, 2019; Shitara et al., 2024; Yaeger et al., 2018). Meanwhile, epidemiological surveys show that left-sided CRC has a higher incidence and better prognosis (Benedix et al., 2010; Price et al., 2015; Shi et al., 2021). On the other hand, for advanced CRC patients, different first-line chemotherapy regimens are potential factors affecting patient prognosis (Yoshino et al., 2024), among which CAPEOX with or without bevacizumab, as a classic first-line chemotherapy regimen for advanced CRC recommended by the Chinese Society of Clinical Oncology guidelines, has been widely used in clinical practice (Diagnosis, Treatment Guidelines For Colorectal Cancer Working Group C, 2019; Yuan et al., 2019). Therefore, in this study, we focus on patients with advanced left-sided CRC who received CAPEOX ± bevacizumab as first-line chemotherapy and explore ways to improve prognosis prediction for this population.

Increasing evidence suggests that the interaction between local immune response and systemic inflammation may play a crucial role in the development and progression of various cancers, including CRC. Neutrophil, lymphocyte, and platelet levels determined by complete blood count may help reveal the systemic inflammatory response (Yan et al., 2020). Since Hu et al. (2014) first described the systemic immune-inflammation index (SII) based on neutrophil, lymphocyte, and platelet counts for predicting the prognosis of hepatocellular carcinoma, studies by Chen et al. (2017) have shown that SII has better prognostic value than the neutrophil-to-lymphocyte ratio (NLR) and platelet-to-lymphocyte ratio (PLR) in CRC patients after radical surgery, and subsequent research by Xie et al. (2018) further confirmed the prognostic value of SII in metastatic colorectal cancer (mCRC). However, under the premise of controlling for treatment regimen and primary tumor location, the comparative prognostic value of different systemic inflammatory indices in advanced CRC remains inconclusive. Therefore, this study aims to evaluate and compare the prognostic value of SII, NLR, PLR, and the monocyte-to-lymphocyte ratio (MLR) in patients with advanced left-sided CRC receiving CAPEOX ± bevacizumab as first-line chemotherapy.

Materials & Methods

Study population

We retrospectively collected data from 231 patients with unresectable advanced (initially unresectable or postoperative recurrent stage IV) left-sided CRC who received CAPEOX ± bevacizumab as first-line chemotherapy at the Oncology Department of the Third Affiliated Hospital of Soochow University from October 2015 to December 2023. The reasons for selecting this patient population are as follows: firstly, a multicenter epidemiological survey from China showed that left-sided colon cancer and rectal cancer accounted for 19.8% and 58.7% of the total incidence, respectively (Shi et al., 2021). Additionally, multiple studies have indicated that, compared to right-sided colon cancer, left-sided colon and rectal cancer have a better prognosis, particularly in stage III–IV patients (Benedix et al., 2010; Price et al., 2015; Weiss et al., 2014). Therefore, these factors highlight the significance of prognostic analysis in patients with left-sided CRC. Secondly, CAPEOX ± bevacizumab, as a classic first-line chemotherapy regimen for advanced CRC recommended by the Chinese Society of Clinical Oncology (CSCO) guidelines, has been widely used in clinical practice (Diagnosis, Treatment Guidelines For Colorectal Cancer Working Group C, 2019; Yuan et al., 2019). Thirdly, although for patients with MSS or MSI-L/p-MMR and wild-type RAS and BRAF, the guidelines recommend first-line chemotherapy with FOLFOX/FOLFIRI ± cetuximab (a two-week regimen), however, for patients with limited financial resources, CAPEOX ± bevacizumab (a three-week regimen) can also be used (Diagnosis, Treatment Guidelines For Colorectal Cancer Working Group C, 2019; Yuan et al., 2019). Therefore, to ensure relative homogeneity in treatment regimens and treatment cycles among patients, we ultimately included patients who received CAPEOX ± bevacizumab as first-line chemotherapy and conducted the corresponding data collection and statistical analysis.

Patients eligible for inclusion met the following criteria: (1) a diagnosis of initially unresectable or postoperative recurrent stage IV metastatic CRC, as defined by the 2024 Chinese Society of Clinical Oncology (CSCO) Guidelines for CRC; (2) histopathological confirmation of adenocarcinoma of the left colon (distal to the splenic flexure) or rectum; (3) eligibility for palliative chemotherapy with CAPEOX ± bevacizumab, administered according to CSCO guidelines; and (4) availability of complete medical records. Patients were excluded if they had incomplete medical records, severe infections or acute inflammatory conditions affecting hematologic parameters, or concurrent primary malignancies at other sites. Ultimately, 231 patients with unresectable advanced (initially unresectable or postoperative recurrent stage IV) left-sided CRC were selected for the study.

This study was conducted in accordance with the principles of the Declaration of Helsinki and was approved by the Ethics Committee of the Third Affiliated Hospital of Soochow University (Approval No.: 2023 J. No. 060, Approval Date: October 27, 2023, Valid until: June 15, 2026). Since some patients have passed away or are difficult to contact due to the passage of time, it is objectively impossible to obtain their informed consent. Therefore, the Ethics Committee approved the waiver of informed consent or the waiver of certain elements of informed consent.

Collection of clinicopathological data

Clinicopathological characteristics and laboratory findings obtained within one month prior to initiating first-line chemotherapy were retrospectively extracted from the electronic medical record system. The variables collected included patient demographic data (age, gender, height and weight), molecular profiles (KRAS/NRAS/BRAF mutation status), and clinical features (presence of liver-confined metastases, resection status of the primary tumor, history of adjuvant chemotherapy). Additionally, data on first-line chemotherapy regimens and hematologic parameters, including neutrophil, platelet, monocyte, and lymphocyte counts, were obtained.

Calculation formulas for systemic inflammatory indices and body mass index

NLR = Neutrophil count/Lymphocyte count, PLR = Platelet count/Lymphocyte count, MLR = Monocyte count/Lymphocyte count, SII = (Neutrophil count × Platelet count)/Lymphocyte count, BMI = weight (kg)/height2 (m2).

Follow-up and outcome assessment

Patients were monitored through a combination of outpatient and inpatient data from the electronic medical record system and telephone follow-ups. OS was defined as the duration from the diagnosis of initially unresectable or postoperatively recurrent stage IV metastatic CRC (as confirmed through imaging or pathological evaluation) to either the date of death or the last follow-up. The study follow-up concluded on October 11, 2024, with durations ranging from 1.0 to 103.2 months and a median follow-up time of 23.6 months. Among the cohort, 26 patients were lost to follow-up, corresponding to a loss-to-follow-up rate of 11.3%.

Statistical methods

Data were organized using Microsoft Excel 2019 (Microsoft, Redmond, WA, USA), while statistical analyses and visualizations were performed using SPSS version 26 (IBM Corp., Armonk, NY, USA) and R version 4.2.3 (R Core Team, 2022). X-Tile version 3.6.1 was used to establish cutoff values for age, body mass index (BMI), NLR, PLR, MLR, and SII levels in predicting OS in advanced left-sided CRC. Based on these thresholds, the study cohort was stratified into high-value and low-value groups.

To minimize the risk of overfitting in small sample sizes when using X-Tile to determine cutoff values based on the log-rank test, we adopted a 2-fold cross-validation approach proposed by Faraggi & Simon (1996) to validate the reliability of the cutoff selection. This method has been applied in recent prospective studies (Chen et al., 2021; Porter et al., 2024; Syeda et al., 2021). The specific steps of this method are as follows: for each variable requiring a cutoff, the entire sample was randomly divided into two subsets, Subset I and Subset II, ensuring each subset contains a similar number of observations. Using Subset I, X-Tile was employed to determine the cutoff value; this cutoff was then applied to Subset II to categorize patients into Group A (above the cutoff) and Group B (below the cutoff). Similarly, a cutoff was determined using Subset II and applied to Subset I. After this process, each patient was assigned to either Group A or Group B based on the cutoff derived from the other subset. A log-rank test stratified by subset was then performed to assess the significance of survival differences between the two groups. If the result was significant, it indicated that the risk of overfitting when directly applying X-Tile to the entire sample was acceptable. Following this analysis, significant survival differences were observed between the two groups for all variables except BMI (Table S1). Therefore, for BMI, we adopted the WHO standard classification for Asian populations. Given the small number of patients in the underweight and obese categories, we referred to the study by Liu et al. (2013) and divided all patients into two groups based on a BMI threshold of 23.0.

Chi-square (χ2) tests were conducted to compare the clinicopathological characteristics of patients across varying levels of NLR, PLR, MLR, and SII. Kaplan–Meier methods were employed for survival analysis, with survival curves generated and compared using log-rank tests. Univariable analysis was performed using the Cox proportional hazards model to identify prognostic factors. To assess the correlations between systemic inflammatory indices (SII, NLR, PLR, MLR), the Shapiro–Wilk test for normality and Spearman’s correlation analysis were employed. To avoid potential collinearity among the systemic inflammatory indices, the indices that were statistically significant in the univariable Cox analysis were separately included in multivariable Cox regression analyses along with the clinicopathological factors that were also statistically significant in the univariable Cox analysis.

Using R version 4.2.3, five-fold cross-validation was conducted by randomly partitioning the dataset into five approximately equal subsets. A fixed random seed was applied to ensure reproducibility. Each subset was alternately designated as the validation set, with the remaining subsets serving as the training set. This process was repeated five times. The average C-index for systemic inflammatory indices identified in multivariable analyses was calculated across the five training and validation sets to evaluate prognostic value. The stability and generalizability of the C-index across different data partitions were further evaluated by calculating the mean squared error (MSE) between the training and validation sets during cross-validation. The formula for MSE is as follows: MSE=1n∑i=1nyi−y ˆ12

where n is the number of cross-validation iterations, and yi and y ˆ1 represent the C-index of the training and validation sets in the i-th cross-validation iteration.

Finally, we employed the Cox proportional hazards model to assess the interaction effect between the SII and treatment regimen in advanced left-sided CRC. We constructed a Cox proportional hazards model that included SII, treatment regimen (CAPEOX versus CAPEOX plus bevacizumab), and their interaction term. By examining the interaction term, we determined whether the prognostic value of SII varied by treatment regimen or resection status. The significance level was set at α = 0.05.

Results

Association between SII and clinicopathological characteristics in patients with advanced left-sided CRC

This study encompassed 231 patients, with a median age of 65 years. Of these patients, 162 (70.1%) were male, and 69 (29.9%) were female. Primary tumor resection had been performed in 137 patients, while 94 patients did not undergo surgical intervention. Regarding first-line chemotherapy, 97 patients were treated with the CAPEOX regimen (86 with initially unresectable disease and 11 with recurrent disease), and 134 received CAPEOX combined with bevacizumab (93 with initially unresectable disease and 41 with recurrent disease). Using the X-Tile 3.6.1 software, optimal cutoff values for predicting OS were established for various parameters, including age (67 years), NLR (2.8), PLR (184.4), MLR (0.2), and SII (1,424.8). Based on the SII cutoff, patients were stratified into low-SII (SII ≤ 1,424.8) and high-SII (SII > 1,424.8) groups. Statistically significant differences were observed in primary tumor resection status between these groups (P < 0.05). Further details regarding the distribution of SII and clinicopathological variables between the two groups are provided in Table 1. Other systemic inflammatory indices are presented in Table S2.

Table 1 Clinicopathological characteristics of patients with advanced left-sided colorectal cancer group by SII.

Clinicopathological characteristics	SII	P-value	
	Low	High		
Gender			0.937	
Male	145	17		
Female	62	7		
Age (years)			0.810	
≤67	126	14		
>67	81	10		
BMI (Kg/m2)			0.164	
≤23	107	16		
>23	100	8		
Treatment regimen			0.364	
CAPEOX	89	8		
CAPEOX+Bev	118	16		
Primary Tumor resection			0.006 *	
No	78	16		
Yes	129	8		
Site of metastasis			0.520	
Limited to liver	65	6		
Not limited to liver	142	18		
RAS/BRAF status			0.115	
Wild-type	45	6		
Mutant	94	15		
Not tested	68	3		
Prior adjuvant therapy			0.321	
No	168	22		
Yes	39	2		
Notes.

* P < 0.05 was considered statistically significant.

Abbreviation SII systemic immune-inflammation index

Association between systemic inflammatory indices and prognosis of patients with advanced left-sided CRC

Kaplan–Meier survival analysis demonstrated that elevated SII, NLR, PLR, and MLR were each significantly associated with reduced OS (all P < 0.05) (Fig. 1). Univariable Cox regression analysis identified age, primary tumor resection, SII, NLR, PLR, and MLR as significant prognostic factors influencing OS (P < 0.05; Table 2).

Figure 1 Kaplan–Meier overall survival curves for patients with advanced left-sided colorectal cancer group by systemic inflammation indices.

P < 0.05 was considered statistically significant. Abbreviations: SII, systemic immune-inflammation index; NLR, neutrophil-to-lymphocyte ratio; PLR, platelet-to-lymphocyte ratio; MLR, monocyte-to-lymphocyte ratio.

Table 2 Univariable and multivariable Cox proportional-hazards analysis of overall survival in patients with advanced left-sided colorectal cancer.

Variables	Univariate cox analysis	Multivariate cox analysis	
	HR	P-value	HR	P-value	
Gender					
Female	Reference				
Male	0.788 (0.535–1.160)	0.227			
Age (years)					
≤67	Reference				
>67	1.621 (1.132–2.320)	0.008 *	1.607 (1.121–2.303)	0.010 a	
BMI (Kg/m2)					
≤23	Reference				
>23	1.069 (0.749–1.525)	0.714			
Treatment Regimen					
CAPEOX	Reference				
CAPEOX+Bev	1.374 (0.950–1.985)	0.091			
Primary Tumor Resection					
No	Reference				
Yes	0.52 (0.363–0.746)	<0.001 *	0.532 (0.370–0.766)	0.001 a	
Site of Metastasis					
Limited to Liver	Reference				
Not Limited to Liver	1.045 (0.716–1.525)	0.820			
RAS/BRAF Status					
Wild-Type	Reference				
Mutant	1.354 (0.810–2.263)	0.248			
Not Tested	1.301 (0.770–2.200)	0.325			
Prior Adjuvant Therapy					
No	Reference				
Yes	0.954 (0.570–1.596)	0.858			
SII					
≤1,424.8	Reference				
>1.424.8	3.788 (2.148–6.679)	<0.001 *	3.534 (1.988–6.282)	<0.001 a	
NLR					
≤2.8	Reference		Reference		
>2.8	2.267 (1.575–3.263)	<0.001 *	1.844 (1.243–2.737)	0.002 b	
PLR					
≤184.4	Reference		Reference		
>184.4	2.021 (1.390–2.940)	<0.001 *	1.707 (1.156–2.520)	0.007 c	
MLR					
≤0.2	Reference		Reference		
>0.2	2.223 (1.465–3.371)	<0.001 *	1.832 (1.191–2.817)	0.006 d	
Notes.

* P < 0.05 was considered statistically significant.

Abbreviations HR hazard ratio

CI confidence interval

SII systemic immune-inflammation index

NLR neutrophil-to-lymphocyte ratio

PLR platelet-to-lymphocyte ratio

MLR monocyte-to-lymphocyte ratio

Multivariate cox models: a, includes SII, patient age, and primary tumor resection; b, includes NLR, age, and primary tumor resection; c, includes PLR, age, and primary tumor resection status; d, includes MLR, age, and primary tumor resection.

To assess the correlations between the SII, NLR, PLR, and MLR, we employed the Shapiro–Wilk test to evaluate the normality of these systemic inflammatory markers. The results demonstrated that all indices deviated from a normal distribution (P < 0.05) (Table S3). Subsequently, a Spearman rank correlation heatmap was constructed, which revealed a moderate degree of correlation among the systemic inflammatory indices, with correlation coefficients ranging from 0.234 to 0.530 (Fig. S1). Therefore, to avoid potential collinearity issues, we conducted multivariable analysis using four separate models. Each multivariable model included only one systemic inflammatory index. Multivariable Cox regression analysis confirmed SII (HR = 3.534, 95% CI [1.988–6.282], P < 0.001), NLR (HR = 1.844, 95% CI [1.243–2.737], P = 0.002), PLR (HR = 1.707, 95% CI [1.156–2.520], P = 0.007), and MLR (HR = 1.832, 95% CI [1.191–2.817], P = 0.006) as independent prognostic factors for OS (Table 2).

Comparison of prognostic value of systemic inflammatory indices

The prognostic value of SII was evaluated in comparison with NLR, PLR, and MLR, utilizing a 5-fold cross-validation approach. The mean C-index for both the training and validation datasets was calculated across the five validation folds, alongside MSE. The findings revealed that SII demonstrated a higher mean C-index relative to NLR, PLR, and MLR in both the training and validation sets. Additionally, SII exhibited lower MSE values, underscoring its superior prognostic value, robustness, and generalizability across different data partitions (Fig. 2; Table 3).

Figure 2 Distribution of C-index in 5-fold cross-validation for systemic inflammation indices.

Abbreviations: SII, systemic immune-inflammation index; NLR, neutrophil-to-lymphocyte ratio; PLR, platelet-to-lymphocyte ratio; MLR, monocyte-to-lymphocyte ratio.

Table 3 Average C-index and MSE of systemic inflammation indices in 5-fold cross-validation.

Variable	Validation set average C-index	Training set average C-index	Mean squared error (MSE)	
NLR	0.6887	0.6996	0.0179	
PLR	0.6718	0.6817	0.0257	
SII	0.7965	0.7981	0.0142	
MLR	0.7089	0.7013	0.0373	
Notes.

C-index, Harrell’s concordance index, quantifies model discriminative ability; MSE, mean squared error, assesses the stability of performance between training and validation sets.

Abbreviations SII systemic immune-inflammation index

NLR neutrophil-to-lymphocyte ratio

PLR platelet-to-lymphocyte ratio

MLR monocyte-to-lymphocyte ratio

Interaction analysis between SII and treatment regimen

Additionally, the interaction effect between treatment regimens (CAPEOX ± bevacizumab) and SII categories (high-value group vs. low-value group) was assessed. Multivariable Cox proportional hazards analysis revealed no statistically significant interaction between the treatment regimens and SII (P = 0.935). Similarly, Kaplan–Meier survival analysis and log-rank tests demonstrated no significant differences in survival outcomes between the CAPEOX and CAPEOX + bevacizumab treatment groups (P = 0.09). These findings suggest that variations in treatment regimens do not significantly modify the prognostic value of SII.

Discussion

In summary, our study found that among patients with advanced left-sided CRC receiving CAPEOX ± bevacizumab as first-line chemotherapy, those with higher pretreatment SII had worse OS. Furthermore, the prognostic value of SII was superior to that of the NLR, PLR, and MLR.

Our study indicates that SII, NLR, PLR, and MLR are independent prognostic factors for OS in CRC patients. Previously, multiple studies on the correlation between systemic inflammatory indices and the prognosis of CRC patients have reached similar conclusions: Li et al. (2023), in a meta-analysis of 18 retrospective cohort studies, demonstrated that elevated preoperative NLR was significantly associated with reduced OS (multivariable HR = 1.83, 95% CI [1.61–2.08], P < 0.01), disease-free survival (DFS) (multivariable HR = 1.78, 95% CI [1.16–2.71], P < 0.01), and recurrence-free survival (RFS) (multivariable HR = 1.46, 95% CI [1.15–1.85], P < 0.01) in CRC patients undergoing surgery for liver metastases. Similarly, a large-scale retrospective analysis by Kim et al. (2017) involving 1,868 CRC patients identified an elevated PLR (≥160) as an independent predictor of poor OS and DFS in stage III and IV CRC patients; however, this association was not observed in stage I and II patients. Furthermore, Jakubowska et al. (2020) reported that postoperative MLR (HR = 2.903, 95% CI [1.368–6.158], P = 0.005) was an independent prognostic indicator for CRC patients undergoing radical resection. The SII, a parameter of increasing clinical interest, has also demonstrated robust prognostic utility. Xie et al. (2018), in a cohort of 240 newly diagnosed stage IV metastatic CRC patients who underwent surgical resection, found that elevated SII was independently associated with diminished OS (HR = 1.462, 95% CI = 1.049–2.038, P = 0.025). Additionally, consistent with the findings of our study, Chen et al. (2017) demonstrated that the SII exhibits superior prognostic value for OS and DFS compared to NLR and PLR in CRC patients following radical surgery. Furthermore, a retrospective analysis of 118 CRC patients prior to radical resection highlighted SII (P = 0.0031) as an independent predictor of tumor recurrence post-surgery, with an area under the curve (AUC) of 0.710, outperforming other inflammatory indices, including NLR, PLR, and MLR. A potential explanation for the superior prognostic utility of SII may lie in its more comprehensive reflection of the inflammation and immune response status compared to other systemic inflammatory indices (Chen et al., 2017; Nakamoto et al., 2023). However, it is noteworthy that the patient cohorts in these studies exclusively comprised individuals who had undergone radical surgery.

However, for patients with advanced CRC, variations in first-line chemotherapy regimens represent potential factors influencing prognosis (Yoshino et al., 2024). Additionally, research has demonstrated that left-sided and right-sided CRC exhibit differences in tissue origin and biological characteristics, which result in variations in drug responsiveness and patient survival outcomes (Piawah & Venook, 2019; Shitara et al., 2024; Yaeger et al., 2018). Consequently, in our study, we evaluated the prognostic value of various systemic inflammatory indices, including the SII, while controlling for primary tumor location and first-line chemotherapy regimen. The findings revealed that multivariable Cox regression analysis indicated a poorer prognosis among patients with elevated SII levels prior to first-line chemotherapy (HR = 3.534, 95% CI [1.988–6.282], P < 0.001). Moreover, the C-index, derived from 5-fold cross-validation across both training and validation sets, demonstrated that SII outperformed the NLR, PLR, and MLR in prognostic value.

It is noteworthy that in recent years, several meta-analyses examining the prognostic value of the SII in CRC have consistently confirmed an association between elevated SII levels and poor prognosis (Dong et al., 2020; Menyhart, Fekete & Győrffy, 2024; Tan et al., 2025). However, the findings of some studies warrant further discussion. For example, while the study by Passardi et al. (2016) validated the prognostic significance of SII, it did not demonstrate any superiority of SII over the PLR and NLR. This discrepancy may be attributed to differences between their study and ours, including the first-line chemotherapy regimens (FOLFOX and FOLFIRI ± bevacizumab), the study population (European cohort), and the lack of control for tumor location. Moreover, studies by Jiang et al. (2019); Yang et al. (2017a) found that, within Asian cohorts composed predominantly of patients with left-sided CRC, the SII was not established as an independent prognostic factor for OS in mCRC patients receiving first-line chemotherapy combined with cetuximab. Additionally, studies on the NLR and PLR have indicated that these indices exhibit more pronounced prognostic value in left-sided CRC, whereas their association is weaker in right-sided CRC (Mazaki et al., 2020; Yang et al., 2017b). Although these investigations did not directly evaluate SII, given its correlation with NLR and PLR, this suggests that primary tumor location may similarly influence the prognostic performance of SII, warranting further exploration. Consequently, the aforementioned studies highlight the necessity of validating the prognostic value of SII across diverse CRC subgroups, such as those defined by ethnicity, treatment regimens, and primary tumor locations. In conclusion, we propose that in future clinical practice, refined prognostic predictions tailored to specific subgroups should be implemented to mitigate the interference of confounding biases.

Interaction and stratified analyses were conducted to investigate the influence of treatment regimens on the prognostic value of SII. Multivariable survival analysis using the Cox proportional hazard model demonstrated no significant interaction between SII and treatment regimens (P = 0.935). Kaplan–Meier survival curves and log-rank tests further indicated no statistically significant difference in survival outcomes between patients receiving CAPEOX and those treated with CAPEOX + bevacizumab (P = 0.09). These findings align with results from a prior phase III randomized clinical trial, which evaluated the efficacy of bevacizumab combined with oxaliplatin-based chemotherapy (CAPEOX or FOLFOX-4) as first-line chemotherapy for metastatic CRC and found no significant difference in OS between the bevacizumab and placebo groups (P = 0.077) (Saltz et al., 2023). These observations may be attributed to differences in subsequent-line treatment strategies.

This study still has some limitations. Firstly, it was restricted to patients with advanced left-sided CRC who received CAPEOX ±  bevacizumab as first-line chemotherapy, which may not reflect a broader patient population undergoing other standard therapies (such as FOLFOX/FOLFIRI ± cetuximab) or with different primary tumor locations, thereby limiting the generalizability of the findings. Additionally, the study employed a small-sample, single-center retrospective design without an external validation cohort, which may introduce selection bias and constrain causal inference. Moreover, potential confounding factors, such as alterations in systemic inflammation due to comorbidities or other treatments, may not have been adequately accounted for. Future research could consider large-scale, multi-center prospective studies with extended follow-up periods and inclusion of patients with right-sided CRC to more comprehensively evaluate the prognostic value of the SII.

Conclusions

Among patients with advanced left-sided CRC receiving CAPEOX ±  bevacizumab as first-line chemotherapy, elevated pre-treatment SII was associated with worse OS, and SII outperformed NLR, PLR, and MLR as a prognostic index. However, the study’s conclusions are limited by its relatively small sample size. Future large-scale, multicenter prospective studies are warranted to further validate the relationship between systemic inflammation and patient outcomes.

Supplemental Information

Supplemental Information 1 Raw data

Supplemental Information 2 5 fold cross validation

Supplemental Information 3 KM curve (R code)

Supplemental Information 4 Spearman correlation heatmap of systemic inflammation indices

Supplemental Information 5 Two-fold cross-validation of X-Tile–defined cutoff values for variables in advanced left-sided colorectal cancer

† Cutoff values were derived in one subset, applied to the other, and tested with a subset-stratified log-rank test on the merged data; P < 0.05 denotes a significant survival difference. Abbreviations: SII, systemic immune-inflammation index; NLR, neutrophil-to-lymphocyte ratio; PLR, platelet-to-lymphocyte ratio; MLR, monocyte-to-lymphocyte ratio; BMI, body mass index.

Supplemental Information 6 Clinicopathological characteristics of patients with advanced left-sided colorectal cancer group by NLR, PLR, and MLR

P < 0.05 denotes a significant survival difference. Abbreviations: SII, systemic immune-inflammation index; NLR, neutrophil-to-lymphocyte ratio; PLR, platelet-to-lymphocyte ratio; MLR, monocyte-to-lymphocyte ratio; BMI, body mass index.

Supplemental Information 7 Shapiro–Wilk test of normality for systemic inflammation indices

* P < 0.05 indicates a significant departure from normality. Abbreviations: NLR, neutrophil-to-lymphocyte ratio; PLR, platelet-to-lymphocyte ratio; MLR, monocyte-to-lymphocyte ratio; SII, systemic immune-inflammation index.

Supplemental Information 8 Spearman correlation heatmap of systemic inflammation indices

Each cell shows the Spearman rank-correlation coefficient (ρ) between two indices; deeper red indicates a stronger positive association (scale −1 to +1). Asterisks denote statistical significance (P < 0.01). Abbreviations: NLR, neutrophil-to-lymphocyte ratio; PLR, platelet-to-lymphocyte ratio; MLR, monocyte-to-lymphocyte ratio; SII, systemic immune-inflammation index.

We express gratitude to the study participants for allowing the use of their data in this research and to Phoebe Chi, MD, from Liwen Bianji (Edanz), for editing a draft of this manuscript.

Additional Information and Declarations

Competing Interests

Author Contributions

Human Ethics

Data Availability

The authors declare there are no competing interests.

Jingyue Su conceived and designed the experiments, performed the experiments, analyzed the data, prepared figures and/or tables, authored or reviewed drafts of the article, and approved the final draft.

Shaokun Yu analyzed the data, authored or reviewed drafts of the article, and approved the final draft.

Yanjie Xu conceived and designed the experiments, authored or reviewed drafts of the article, and approved the final draft.

Jiemin Zhao conceived and designed the experiments, authored or reviewed drafts of the article, and approved the final draft.

Wenwei Hu conceived and designed the experiments, authored or reviewed drafts of the article, and approved the final draft.

Xuefeng Ni conceived and designed the experiments, performed the experiments, authored or reviewed drafts of the article, and approved the final draft.

The following information was supplied relating to ethical approvals (i.e., approving body and any reference numbers):

This study adhered to the principles outlined in the Declaration of Helsinki and was approved by the Ethics Committee of the Third Affiliated Hospital of Soochow University (Approval Number: 2023 J. No. 060). Ethical approval included a waiver of informed consent or a partial waiver of consent elements, as appropriate.

The following information was supplied regarding data availability:

The raw data and code are available in the Supplemental Files.

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
