# Peer review of "Predictive role of systemic immune-inflammation index in the prognosis of patients with advanced left-sided colorectal cancer: a retrospective study"

_PeerJ, doi:10.7717/peerj.20095_

## Round 0.1 · original submission · Major Revisions

Please address all three reviews.

Reviewer 1 ·

Basic reporting

The manuscript meets basic reporting standards, with clear language, structured content, and well-referenced literature. Minor improvements in sentence clarity and additional citations on SII and related biomarkers would further enhance its quality.

Experimental design

The study has several limitations, primarily due to its retrospective design, which may introduce selection bias and limit causal interpretations. The sample size (231 patients), while informative, is relatively small, and the absence of an external validation cohort reduces the robustness of the prognostic conclusions. Additionally, the study focuses exclusively on left-sided colorectal cancer, limiting its applicability to right-sided CRC, which has distinct biological and clinical characteristics. Although Cox regression analysis was performed, potential confounders, such as variations in systemic inflammation due to comorbidities or other treatments, may not be fully accounted for. Future research should include prospective validation, larger and more diverse cohorts, and consideration of right-sided CRC patients to enhance generalizability.

Validity of the findings

The findings are statistically robust, supported by Kaplan-Meier survival analysis, Cox regression models, and 5-fold cross-validation. The Systemic Immune-Inflammation Index (SII) is demonstrated as a superior prognostic marker compared to NLR, PLR, and MLR, with a higher C-index and lower MSE. However, the lack of an external validation cohort and potential confounding factors (e.g., inflammation from non-cancer conditions) may limit the strength of the conclusions. While the results suggest clinical utility, further prospective validation is needed to confirm reproducibility and generalizability.

Reviewer 2 ·

Basic reporting

This research investigated the prognostic significance of the systemic immune inflammation index (SII) in patients with advanced colorectal cancer. The findings revealed that in individuals with advanced left-sided CRC, elevated pre-treatment SII was associated with poorer prognoses. Furthermore, SII demonstrated superior predictive performance compared to neutrophil-to-lymphocyte ratio (NLR), platelet-to-lymphocyte ratio (PLR), and monocyte-to-lymphocyte ratio (MLR). Increased pre-treatment SII values were also correlated with poorer survival outcomes, emphasizing the clinical utility of SII in prognostic stratification.

Professional English was ok, but lacked flow.

In terms of references, I was surprised that one of the biggest studies analyzing pre-treatment inflammation and immune-based scores as predictors of treatment efficacy in mCRC to date, published by Passardi et al., was not even referenced.

Experimental design

The experimental design/stat analysis looks ok.

Please explain the SSI cutoff and stratification of low vs high SSI as 1424.8
Could you do a multivariate analysis to elucidate which factors are important for survival improvement? Is the prior number of lines important? Is the CEA level important?

Validity of the findings

I also feel the authors need to differentiate between prognostic value and predictive value. A "prognostic value" refers to the ability of a factor to predict the likely course or outcome of a disease regardless of treatment, while a "predictive value" indicates how likely a patient is to respond favorably to a specific treatment.

Additional comments

This was a very hard paper to follow because of the way it is written. The authors need to rewrite this paper in a manner that is intuitive (especially the results section)- with the authors' interpretations of the results, and what its impact may be.

I recommend that the authors consider professional editing to improve the manuscript's flow and readability. Additionally, the discussion section would benefit from a more structured approach and a thorough comparison of the present findings with existing literature.

Reviewer 3 ·

Basic reporting

The research article titled "Predictive role of Systemic Immune-Inflammation Index (SII) in the prognosis of patients with advanced left-sided colorectal cancer" investigates the prognostic value of SII in patients undergoing CAPEOX ± bevacizumab therapy. The study retrospectively analyzed 231 patients, identifying SII as a robust prognostic marker that outperforms other systemic inflammation indices, such as NLR, PLR, and MLR. The findings suggest that elevated pre-treatment SII values are associated with poorer survival outcomes, emphasizing their clinical utility in prognostic stratification. However, the study focuses exclusively on patients treated with CAPEOX ± bevacizumab, which may not reflect the broader population of advanced colorectal cancer patients receiving other standard treatments, such as FOLFOX/FOLFIRI ± cetuximab. This limits the generalizability of the findings to other treatment regimens. While the study is promising, it has several issues that need to be addressed:

1. The introduction is too brief and lacks sufficient detail. No general information about colorectal cancer is provided. I suggest moving some sections from the discussion (lines 227–236 and 279–286) to the introduction and shortening them accordingly. Additionally, the rationale for focusing on left-sided colorectal cancer and SII is not clearly articulated.
2. The discussion section is underdeveloped. Many sections would be more appropriate in the introduction (lines 227–236 and 279–286), methods (lines 254–278), or results (lines 287–295). The discussion does not sufficiently explore the implications of the findings or compare them with existing literature. Furthermore, it does not adequately address how the findings could influence clinical decision-making or patient management, reducing the study's practical impact.
3. The figures and tables lack sufficient clarity. Their legends are not descriptive enough to make them self-explanatory.
4. The manuscript contains several grammatical errors and awkward phrasing. The English language should be improved to ensure clarity for an international audience. Additionally, the terminology is inconsistent, e.g., "Systemic Inflammation Indices" and "Systemic Inflammation Indicators." I recommend having the manuscript reviewed by someone proficient in English and familiar with the subject matter or using a professional editing service.
5. Thank you for providing raw data and R codes. However, the manuscript lacks a "Data and Materials Sharing" section. Please include this section and specify that the data is provided in the supplementary material.
6. The manuscript does not fully conform to PeerJ standards. For example, according to PeerJ guidelines, use "Fig. 1" and "Table S1" instead of "Figure 1" and "Supplementary Table 1," respectively. Please, be sure to double-check the formatting guidelines.
7. The manuscript does not include a funding statement. How was the study funded?

Experimental design

8. The study does not adequately highlight the specific knowledge gap. Please clarify the benefit of your study to the existing literature. The statement in the introduction that "this study will evaluate the potential impact of different treatment regimens on the prognostic relevance of these markers" is confusing, as this is a retrospective study. Can the authors clarify this?
9. The methods are not described with sufficient detail. For example, the rationale for using specific cut-offs for SII, NLR, PLR, and MLR is not well explained. While the use of X-Tile software is mentioned, it is not elaborated upon. Why was X-Tile software used to determine optimal cut-off values instead of median cut-offs, which are widely used in survival analyses? Using X-Tile can lead to overfitting, especially given the unequal group sizes and the small number of individuals in the high-SII group. I suggest either using median cut-off values or comparing your results with those obtained using median cut-offs. Alternatively, validate your results in an independent dataset, ideally an external cohort. Your cohort could also be divided into training and test sets, with cut-off values determined in the training set and validated in the test set.
10. If the multivariable analyses included several systemic markers simultaneously, please comment on the interpretability of the models, as these markers are not independent variables.
11. Although ethical approval is mentioned, the justification for waiving informed consent is not adequately explained, which could raise concerns about the ethical handling of patient data. Additionally, the attached ethical approval appears to have expired. Please provide an updated approval.

Validity of the findings

Underlying data have been provided, but not all analyses are clearly explained.

Additional comments

Minor comments:
Abstract:
12. The objective does not mention "left-sided" colorectal cancer, even though it is stated in the title.
13. Incorrect phrasing. Systemic inflammatory markers were calculated, and blood counts were collected.
14. Revise the sentence: "Survival outcomes were analyzed with the Kaplan-Meier method, while the Cox proportional hazards model identified systemic inflammatory markers associated with survival."
15. Use "univariable" and "multivariable" analyses instead of "univariate" and "multivariate," as the latter terms are incorrect for the analyses performed. Repeats throughout the manuscript.
Introduction:
16. The sentence "These findings underscore the potential interplay between systemic inflammation and the tumor inflammatory microenvironment" (lines 49–50) is imprecise and lacks sufficient evidence. Systemic inflammatory markers, such as CRP, are nonspecific and may not correlate with tumor tissue characteristics.
17. Reformulate the statement about systemic inflammation markers being "well-established prognostic indicators" (line 51).
18. Replace or remove the words "comparative" (line 53) and "conventional" (line 56).

Methods:
19. The statement "This retrospective study analyzed data from 556 patients" (line 64) is incorrect, as only 231 patients were analyzed.
20. Remove the second and third paragraphs (lines 69–81), as they are confusing and add little value.
21. Clarify at the beginning of the section and line 89 that the 231 patients included also recurrent cases, as mentioned in the criteria (lines 82–83).
22. Specify that BMI was calculated from height and weight and not collected (line 98).
23. Add explanatory sentences for the formulas (lines 105–106).
24. Be more specific about the models and interaction assessments (lines 138–140).

Results:
25. Clarify how many patients in the CAPEOX and CAPEOX + bevacizumab groups had recurrent versus unresectable cancer.
26. Is the genetic profiling mentioned relevant to the study? If so, the authors should elaborate and state how many of these were resected and unresected.
27. Address the potential confounding effects of primary tumor resection and treatment regimen on systemic inflammatory markers (Table 1).
28. Explain why the statement ‘Given the substantial intercorrelations observed among SII, NLR, PLR, and MLR….’ (lines 168–169), as the authors have not really assessed the correlation of the ratios.
29. It is stated that ‘Ongoing follow-up is needed to further evaluate disease progression and subsequent-line therapeutic strategies for these patients.’ It is unclear what the authors mean by that. Please, reformulate.

Discussion:
30. Clarify whether the authors mean "predictive marker" instead of "prognostic accuracy for treatment efficacy" (line 286).
31. Revise "Multivariate Cox proportional hazard analysis" to "Multivariable survival analysis using the Cox proportional hazard model" (line 292).

Conclusions:
32. Remove the last sentence about systemic inflammatory markers reflecting insensitivity to antitumor therapies, as it is speculative and unsupported.

Acknowledgements:
33. Acknowledge the study participants for allowing their data to be used for this study.

---

## Round 0.2 · Minor Revisions

Please address the remaining comments, especially those concerning ethical approval.

Reviewer 2 ·

Basic reporting

N/A

Experimental design

N/A

Validity of the findings

N/A

Additional comments

Satisfied with the rebuttal and the changes made to the manuscript

Reviewer 3 ·

Basic reporting

Thank you for the thoroughly revised manuscript. I am quite satisfied with all the revisions, although the authors should further improve the English language and make sure they are consequent in using the same terminology throughout the manuscript.

Experimental design

I am unfortunately not satisfied with the author’s response on my comment concerning the ethical approval for this study (see below). Although the authors state that all analyses were performed before the ethical approval expired (October 27, 2024), I am hesitant to believe that, as for example, the authors performed several additional analyses during the revision of their manuscript this spring (when the study’s ethical approval was already expired). This is described in the author’s response letter. Therefore, I disagree with the author’s statement that ‘all research activities (including data extraction, analysis, and follow-up) were conducted within the validity period of the ethical approval and that the study complies with ethical requirements.’ Updated approval is necessary and I insist that they must obtain a new ethical approval prior the study can be published.

Comment:
Although ethical approval is mentioned, the justification for waiving informed consent is not adequately explained, which could raise concerns about the ethical handling of patient data. Additionally, the attached ethical approval appears to have expired. Please provide an updated approval.

Response:
We sincerely thank you for pointing out the insufficient explanation in the original manuscript regarding the justification for waiving informed consent. This study is a retrospective analysis, and all data were sourced from inpatient medical records of the Oncology Department at the Third Affiliated Hospital of Soochow University, covering the period from October 2015 to December 2023. The raw data have been de-identified to protect patient privacy. Due to the fact that some patients have passed away or are difficult to contact because of the passage of time (e.g., early cases dating back over eight years), it was objectively impossible to obtain their informed consent. To address this, we submitted an "Application for Waiver of Informed Consent or Waiver of Certain Elements of Informed Consent" during the ethical approval process, which was subsequently approved by the Ethics Committee of the Third Affiliated Hospital of Soochow University (Approval No.: 2023 J. No. 060, Approval Date: October 27, 2023).
This approach complies with both international and national ethical standards, such as the Declaration of Helsinki and China’s Regulations on Ethical Review of Biomedical Research Involving Human Subjects (National Health Commission), which permit the waiver of informed consent in retrospective studies using de-identified data with minimal risk. For reference, please see: https://www.gov.cn/zhengce/2016-10/12/content_5713806.htm. Additionally, we have provided the "Application for Waiver of Informed Consent or Waiver of Certain Elements of Informed Consent" submitted during the ethical approval process as supplementary material (see Supplementary File).
Furthermore, we appreciate your attention to the validity period of the ethical approval. We understand that the reviewer may have concerns due to the current date (April 16, 2025) being later than the date on the attached ethical approval. We clarify that the ethical approval was granted on October 27, 2023, and is valid for one year (until October 27, 2024). The patient follow-up and data collection for this study were completed by October 11, 2024, and all research activities (including data extraction, analysis, and follow-up) were conducted within the validity period of the ethical approval. Therefore, the study complies with ethical requirements, and no updated approval is necessary.
To further clarify this, we have added a detailed description in the methods section of the revised manuscript (lines 115–121):
“This study was conducted in accordance with the principles of the Declaration of Helsinki and was approved by the Ethics Committee of the Third Affiliated Hospital of Soochow University (Approval No.: 2023 J. No. 060, Approval Date: October 27, 2023, Valid until: October 27, 2024). Since some patients have passed away or are difficult to contact due to the passage of time, it is objectively impossible to obtain their informed consent. Therefore, the Ethics Committee approved the waiver of informed consent or the waiver of certain elements of informed consent.”
We confirm that the validity period of the ethical approval aligns with the timeline of the research activities and complies with the requirements of international journals and national ethical regulations.

Validity of the findings

no comment

---

## Round 0.3 · accepted · Accept

The authors have satisfactorily addressed the comments of the Reviewers.

Reviewer 3 ·

Basic reporting

I am satisfied with the changes made to the manuscript.

Experimental design

Thank you for providing updated ethical approvals, which are now valid for the whole period of the study.

Validity of the findings

-